# Thermal Titration Molecular Dynamics (TTMD): Not Your Usual Post-Docking Refinement

**DOI:** 10.3390/ijms24043596

**Published:** 2023-02-10

**Authors:** Silvia Menin, Matteo Pavan, Veronica Salmaso, Mattia Sturlese, Stefano Moro

**Affiliations:** Molecular Modeling Section (MMS), Department of Pharmaceutical and Pharmacological Sciences, University of Padova, Via F. Marzolo 5, 35131 Padova, Italy

**Keywords:** docking, refinement, rescoring, molecular dynamics, protein-ligand interaction fingerprints, thermal titration molecular dynamics, TTMD

## Abstract

Molecular docking is one of the most widely used computational approaches in the field of rational drug design, thanks to its favorable balance between the rapidity of execution and the accuracy of provided results. Although very efficient in exploring the conformational degrees of freedom available to the ligand, docking programs can sometimes suffer from inaccurate scoring and ranking of generated poses. To address this issue, several post-docking filters and refinement protocols have been proposed throughout the years, including pharmacophore models and molecular dynamics simulations. In this work, we present the first application of Thermal Titration Molecular Dynamics (TTMD), a recently developed method for the qualitative estimation of protein-ligand unbinding kinetics, to the refinement of docking results. TTMD evaluates the conservation of the native binding mode throughout a series of molecular dynamics simulations performed at progressively increasing temperatures with a scoring function based on protein-ligand interaction fingerprints. The protocol was successfully applied to retrieve the native-like binding pose among a set of decoy poses of drug-like ligands generated on four different pharmaceutically relevant biological targets, including casein kinase 1δ, casein kinase 2, pyruvate dehydrogenase kinase 2, and SARS-CoV-2 main protease.

## 1. Introduction

Until the 1980s, the discovery of new drugs was mainly related to empirical observations about the symptoms-relief properties of natural products or serendipity [1,2]. Nowadays, thanks to the high availability of experimentally determined structures for pharmaceutically relevant biomolecules, the drug discovery process shifted towards a more rational approach [3]. Specifically, novel therapeutic entities are usually designed by taking into consideration the topological features of the interacting partner, in a process defined as “structure-based drug design” [4].

Alongside advancements in the field of biophysical approaches for the experimental determination of protein-ligand complex structures, a pivotal role in the last two decades has been played by computational methodologies [5]. These “in silico” approaches not only flank experimental techniques in rationalizing the mechanism of action of active molecules but are routinely used as a less expensive and quicker alternative in the earliest stages of drug discovery pipelines [6,7].

Among these methods, one of the most widely adopted is certainly molecular docking, a program that is used by medicinal chemists to predict the most plausible ligand accommodation within the active site of a target biomolecule [8]. Molecular docking programs are composed of two main parts: a search algorithm, which is responsible for exploring the ligand’s conformational degrees of freedom and generating a series of possible ligand orientations within the binding site, and a scoring function, which numerically estimates the quality of protein-ligand interactions for poses generated by the search algorithm [9].

Although search algorithms are usually very proficient in exploring the conformational space available to the ligand, docking can sometimes produce inaccurate results, due to the limitations of scoring functions [10]. For instance, there is a natural variance in docking performances across different biological targets, since each binding site has its peculiar interaction features, which may differ from the ones included in the training set, thus leading to an incorrect pose ranking [11].

To mitigate these known issues of molecular docking, several refinement protocols have been proposed throughout the years [12]. A first possible strategy is to filter docking poses based on prior knowledge of the binding features that are required for potent and selective binding to the target [13]. This approach can be carried out by creating a pharmacophore model or a protein-ligand interaction fingerprint based on the binding pattern of known binders, i.e., true active compounds, and then retaining only those poses which match this set of features [14] Although this strategy is successful in diminishing the false positive rate in docking calculations, it is not a first-principle method; therefore, its usage is limited to those cases where the target and its interaction features have already been characterized [15].

Another approach is to rescore docking poses with more computationally demanding methods such as the molecular mechanics/generalized Born surface area (MM/GBSA) method for free binding energy determination [16]. This method is usually a good compromise between rapidity of execution and quality of calculations, and it can be a useful tool for many cases. Nevertheless, its accuracy is limited by some factors, such as the implicit consideration of solvent molecules and the sensitivity towards the polarity and charge properties of both the ligand and the binding site [17,18].

Finally, another commonly adopted strategy is to use short-scale molecular dynamics simulations to evaluate the time-dependent pose stability [19]. This approach has the advantage of considering both the protein flexibility and the explicit treatment of solvent, which are usually neglected in most docking calculations [20,21]. Although in principle the MD-based post-docking refinement seems like a sound method to distinguish poses based on their persistence, two aspects can impair its usefulness. First, the short timescale that is used for most applications (tens of nanoseconds) is insufficient to sample the unbinding event for most poses, flattening the difference between the native pose and wrong but stable enough decoys [22]. Furthermore, in most cases, simple geometric descriptors such as the ligand RMSD or the center-of-mass distance between the ligand and the protein are used to monitor the pose evolution throughout the simulation, for which there is no clear threshold value to define the unbinding event, other than being heavily influenced by the geometry of the binding site and the ligand structure [23,24].

In the current work, we present the first application of Thermal Titration Molecular Dynamics (TTMD) as an alternative MD-based post-docking refinement tool. Recently developed as an enhanced sampling MD protocol for the qualitative estimation of protein-ligand unbinding kinetics [25], TTMD evaluates the persistence of the native binding mode in a certain protein-ligand complex of interest by combining a series of short MD simulations performed at progressively increasing temperatures with a scoring function based on protein-ligand interaction fingerprints [26]. The protocol has been applied to twelve different protein-ligand complexes from four different pharmaceutically relevant biological targets, including protein kinase CK1δ, protein kinase CK2, pyruvate dehydrogenase kinase 2, and SARS-CoV-2 main protease.

## 2. Results

To extend the applicability domain of the TTMD protocol as a post-docking refinement tool, 12 different test cases on four different pharmaceutically relevant targets of interest for our laboratory were investigated: CK1δ, CK2, PDK2, and M^pro^. Test cases were chosen coherently with the original work, picking up two “tight binders” (IC_50_/K_d_ values in the low nanomolar range) and one “weak binder” (IC_50_/K_d_ above the low micromolar threshold). For each biomolecular target, three different protein-ligand complexes available on the Protein Data Bank were chosen. A list of each system investigated in this work can be found in Table 1, while details about ligands’ physicochemical properties (Appendix A) and relevant statistics for each simulated system (Appendix A) can be found in the Appendix A.

At first, for each protein-ligand system, a self-docking experiment with the PLANTS docking software was executed. The first five docking poses according to the ChemPLP scoring function were used as a starting point for the TTMD simulations. For each investigated docking pose, five independent TTMD replicates were performed. The results for each test case are reported separately and discussed aggregately afterward in the Discussion section.

### 2.1. Protein Kinase CK1δ

The first test case regarded protein kinase 1 δ, a serine-threonine kinase belonging to the family of CK1 kinases [37] that is involved in several neurodegenerative illnesses like Alzheimer’s disease, Parkinson’s disease, and amyotrophic lateral sclerosis [38] due to its pleiotropic nature [39]. The result of the TTMD refinement carried out on docking poses generated for each CK1δ protein-ligand complex is summarized in Figure 1, Figure 2 and Figure 3, while detailed analysis about representative replicates (Appendix A), and per-replicate reports (Appendix A) can be found in the Appendix A. For reference, a comparison between the representative TTMD trajectory for the best (pose 1, MD4), and worst (pose 5, MD5) scoring pose for complex 5MQV, according to the MS coefficient, is highlighted in Video S1.

As can be seen in Figure 1, which summarizes the results of TTMD refinement performed on docking poses of ligand 0CK (PDB ID: 3UZP), although poses are very similar and plausible, the MS value correctly identifies the most correct/native-like poses among the five ones. Particularly, poses 1 and 2, which are practically indistinguishable from a binding pattern perspective (both orientations of the 2-aminopyrimidine moiety ensure a bivalent hydrogen bond interaction with the hinge region of the kinase, while the rest of the molecule is superimposable) have the lowest MS value, which is comparable to the one of the experimentally determined binding mode, coherently with the low RMSD value to the reference. On the contrary, poses 3 and 4, which insert the cyclohexyl moiety within the hydrophobic “selectivity pocket” located behind the hinge region, in the inner-most part of the binding site, present a higher MS value, indicating a lower persistence of the native binding mode. Intriguingly, pose 5, which does not show any interaction with the hinge region, has a better MS value compared to poses 3 and 4: this can be explained by the fact that, due to the conserved orientation compared to poses 1 and 2, pose 5 can quite easily interconvert into pose 2 by a slight rotation towards the hinge region, gaining the directional interaction with the hinge without losing any hydrophobic interaction provided by the rest of the molecule.

A similar trend compared to the first case can also be observed for ligand D5Q (PDB ID: 5MQV). As reported in Figure 2, poses 1–2, which only differ for the relative orientation of the solvent-exposed dimethoxyphenyl-pyrrole moiety, present the lowest MS value, similar to the reference. Pose 3, which is also quite similar to the reference and the first two poses, also displays a good MS coefficient, albeit slightly higher than the first two poses. A possible explanation for this phenomenon is that, although the relative orientation of the dimethoxyphenyl-pyrrole portion of the ligand is similar to the reference, contributing to a marginally lower RMSD value compared to pose 2, the 2-carboxamide-pyridin core, which provides the bidentate interaction with the hinge region, is mildly shifted away from the hinge. A little rearrangement of the ligand core through the course of the simulation allows interconversion into a reference-like pose, accompanied by an alteration of interaction pattern, which explains the reason why the pose presents good stability but with lesser conservation of its initial binding pattern. As already seen for ligand 0CK, in this case also the MS score penalizes poses that do not insert an aromatic ring within the hydrophobic selectivity pocket, despite having an optimal interaction with the hinge (pose 4). Finally, the MS score heavily penalizes pose 5, which does not retain any interaction found in the reference.

**Figure 2 ijms-24-03596-f002:**
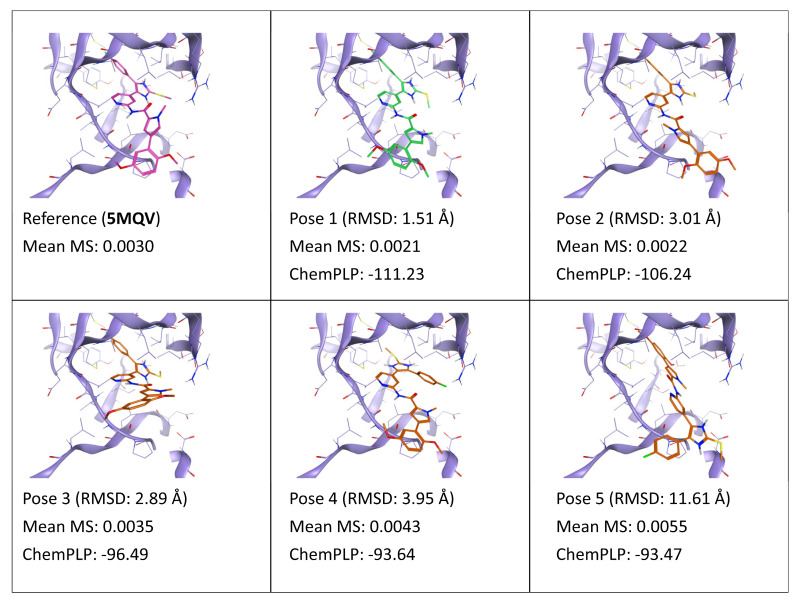
This panel encompasses the results of the TTMD post-docking refinement of poses generated for ligand D5Q within the catalytic site of the protein kinase CK1δ (PDB ID: 5MQV). Protein residues within 5 Å of the ligand (orange) position are shown (violet). For each pose, the RMSD in comparison with the reference, the average MS coefficient resulting from TTMD simulations, and the ChemPLP docking score are reported. As a reference, the experimentally determined binding mode (magenta) is also reported, as well as the MS coefficient derived from its TTMD refinement, while the best docking pose (according to the RMSD to the reference) is highlighted in green.

Differently from the first two cases, involving “tight binding” ligands, the results of the TTMD refinement were inconclusive for ligand AUG (PDB ID: 5IH6). As can be deducted from Figure 3, the best scoring conformation according to the MS value is pose 2, which presents a reasonable although incorrect binding mode, characterized by three hydrogen bonds with the hinge region. This orientation of the triamine-pteridine core, although allowing for a comparable interaction pattern with the hinge, leads to a suboptimal burial of the hydrophobic core of the molecule within the internal portion of the pocket, especially concerning the bromophenyl moiety. However, in this case, the TTMD refinement is not able to discriminate between the native binding mode and other incorrect ones, as noticeable by the fact that the crystal reference and pose 1, which are practically superimposable, present a high MS value, comparable with pose 3, which is incorrect. Intriguingly, as denoted by Appendix A (Appendix A), throughout the simulation, pose 1 interconverts into pose 2, which is the final state of the simulation. Considering that the two poses have a similar quality of interaction with the target, it is possible that the two conformations are in equilibrium and that kinetic factors influence the shift towards either of the two. Nevertheless, another plausible explanation for this phenomenon could be that the temperature ramp utilized, which was the same as the one employed in the original publication to discriminate between strong and weak binders, is too aggressive for ligands with low affinity for the targets. The aggressiveness of the ramp, which was designed to facilitate the observation of unbinding events, could flatten the differences between different poses of the same ligand when the crystal pose itself is unstable in those conditions. If this is the case, a different ramp for this category of binders would have to be developed.

**Figure 3 ijms-24-03596-f003:**
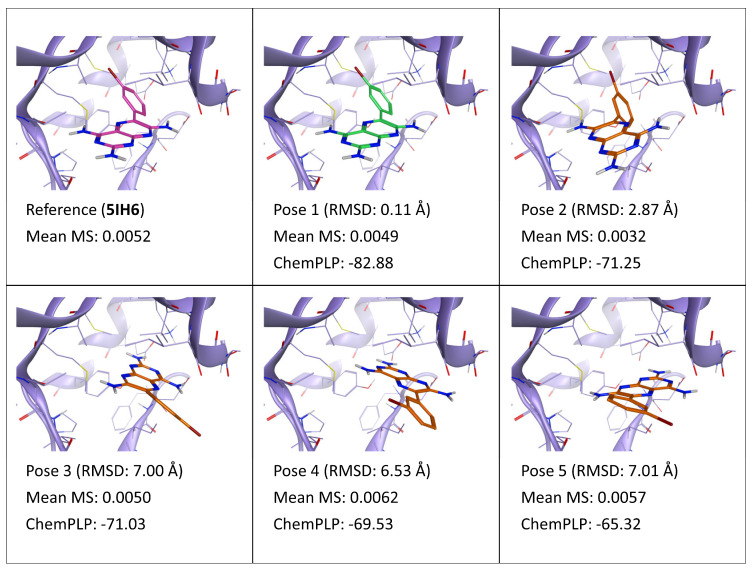
This panel encompasses the results of the TTMD post-docking refinement of poses generated for ligand AUG within the catalytic site of the protein kinase CK1δ (PDB ID: 5IH6). Protein residues within 5 Å of the ligand (orange) position are shown (violet). For each pose, the RMSD in comparison with the reference, the average MS coefficient resulting from TTMD simulations, and the ChemPLP docking score are reported. As a reference, the experimentally determined binding mode (magenta) is also reported, as well as the MS coefficient derived from its TTMD refinement, while the best docking pose (according to the RMSD to the reference) is highlighted in green.

### 2.2. Protein Kinase CK2

The second test case regarded protein kinase 2, a serine-threonine kinase that represents one of the first identified protein kinases [40]. Similar to CK1, thanks to its pleiotropic nature [41] this kinase is involved in several illnesses such as different types of cancer, various neurodegenerative diseases, and viral infections [42]. The result of the TTMD refinement carried out on docking poses generated for each CK2 protein-ligand complex is summarized in Figure 4, Figure 5 and Figure 6, while detailed analysis about representative replicates (Appendix A), and per-replicate reports (Appendix A) can be found in the Appendix A. For reference, a comparison between the representative TTMD trajectory for the best (pose1, MD3) and worst (pose4, MD4) scoring pose for complex 3PE1, according to the MS coefficient, is highlighted in Video S2.

As highlighted by Figure 4, which synthesized the results of the TTMD refinement for ligand 3NG (PDB ID: 3PE1), a similar trend to what has been observed for CK1δ can also be found. Pose 1, which is superimposable to the crystal reference, is associated with the lowest MS value, which is practically identical to the one of the reference. Pose 2, which differs from pose 1 for a slight shift of the naphthyridine core away from the hinge region of the kinase and for the orientation of chlorine atom on the 3-chlorophenyl substituent, displays the second lowest MS value overall, coherently with its suboptimal but overall native-like orientation within the binding site. Pose 3 is penalized by the MS score due to the loss of the pivotal network of direct and water-mediated interactions involving the ligand carboxylate and residues deeply buried in the inner portion of the binding site such as Lys68 and Asp175. Similarly, pose 4 is also heavily penalized by the TTMD-derived score for the blatantly incorrect ligand conformation within the active site: in this case, interactions involving the carboxylate are lost, but the whole naphthyridine core is misplaced. Finally, pose 5 shows an intermediate MS score between the native-like poses and the wrong ones: a possible interpretation for this is that the naphthyridine core of the molecule is deeply buried in a hydrophobic, solvent-shielded area of the binding site, thus compensating for the lack of directional interactions provided by the carboxylate.

**Figure 5 ijms-24-03596-f005:**
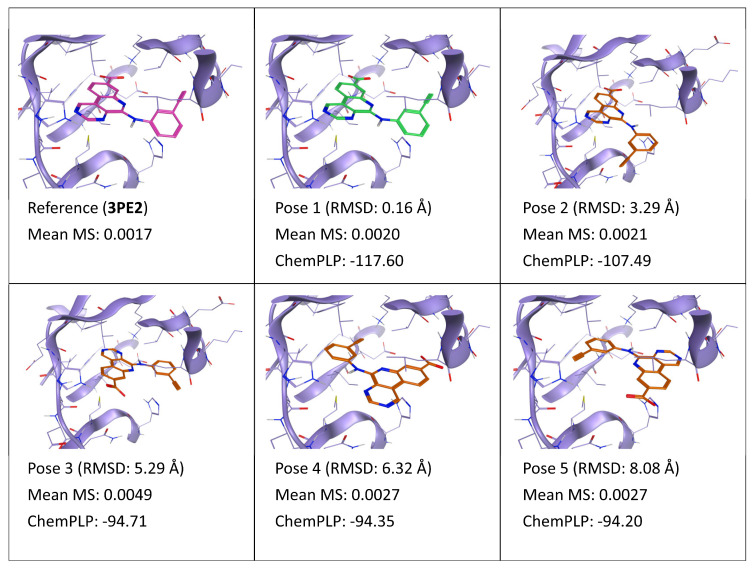
This panel encompasses the results of the TTMD post-docking refinement of poses generated for ligand E1B within the catalytic site of the protein kinase CK2 (PDB ID: 3PE2). Protein residues within 5 Å of the ligand (orange) position are shown (violet). For each pose, the RMSD in comparison with the reference, the average MS coefficient resulting from TTMD simulations, and the ChemPLP docking score are reported. As a reference, the experimentally determined binding mode (magenta) is also reported, as well as the MS coefficient derived from its TTMD refinement, while the best docking pose (according to the RMSD to the reference) is highlighted in green.

As illustrated in Figure 5, results for the TTMD refinement in the case of ligand E1B (PDB ID: 3PE2) recall the ones for ligand 3NG. This is not unexpected, due to the similarities between the two ligands. Particularly, pose 1, which is identical to the reference, presents the lowest MS value possible, which is very similar to the value calculated for the experimentally determined binding mode. The same is true also for pose 2, which only differs for the orientation of the ethynylphenyl substituent but has the same binding pattern for the pyrimidoquinoline core. On the contrary, pose 3, which locates its core near the hinge region of the kinase, as for poses 1 and 2, but presents an incorrect orientation of the carboxylate, leading to the loss of crucial directional interactions with Lys68/Asp175, is linked with a higher MS value. Intriguingly, poses 4 and 5, which are the most different from the reference, present an intermediate MS score, not much higher than the one of native-like poses. A putative explanation for this is that these compounds, 3NG and E1B, both present very rigid and planar structures, enhancing the shape complementarity with the active site and increasing the persistence of short-range hydrophobic and dipolar interactions. Accordingly, despite the lack of directional interactions with the target, both poses 4 and 5 display a good shape and electrostatic complementarity with the pocket, as also indicated by the docking score which is practically identical to that of pose 3, which contributes to the high persistence of the pose despite the lack of the directional interactions provided by the carboxylate.

**Figure 6 ijms-24-03596-f006:**
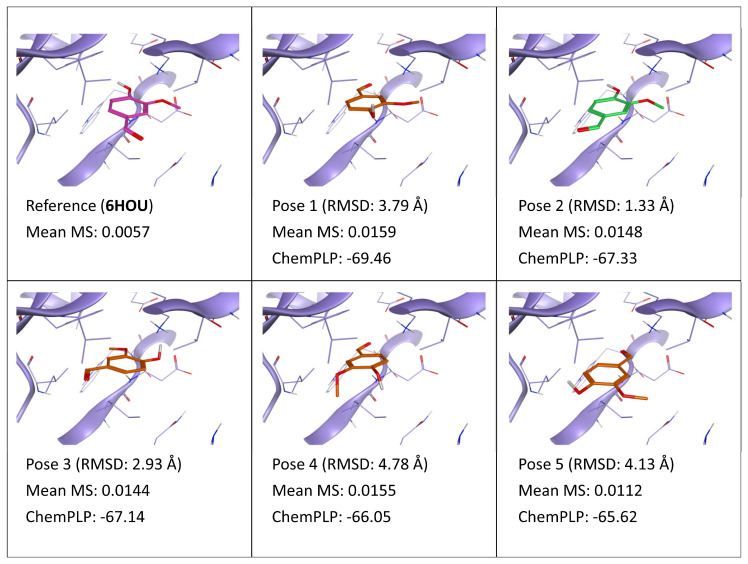
This panel encompasses the results of the TTMD post-docking refinement of poses generated for ligand V55 within the catalytic site of the protein kinase CK2 (PDB ID: 6HOU). Protein residues within 5 Å of the ligand (orange) position are shown (violet). For each pose, the RMSD in comparison with the reference, the average MS coefficient resulting from TTMD simulations, and the ChemPLP docking score are reported. As a reference, the experimentally determined binding mode (magenta) is also reported, as well as the MS coefficient derived from its TTMD refinement, while the best docking pose (according to the RMSD to the reference) is highlighted in green.

In agreement with the results of the TTMD refinement of the CK1δ weak binder, results are inconclusive also for CK2 homologous, ligand V55 (PDB ID: 6HOU). As observable in Figure 6, the MS value is very high for each of the five docking poses, as it is for the reference. High MS values, such as those obtained for V55 poses, indicate that the ligand has rapidly and completely lost its native binding mode in all cases, as also confirmed by detailed trajectory analyses reported in Appendix A (Appendix A). This reinforces the hypothesis that the default temperature ramp is too aggressive to be useful in the refinement of poses for weak binders.

### 2.3. Pyruvate Dehydrogenase Kinase 2

The third test case involved pyruvate dehydrogenase kinase 2, a member of the GHKL ATPase/kinase superfamily that has a central control role upon cellular energy metabolism [43,44] and is involved both in metabolic and neoplastic diseases [45]. The result of the TTMD refinement carried out on docking poses generated for each PDK2 protein-ligand complex is summarized in Figure 7, Figure 8 and Figure 9, while detailed analysis about representative replicates (Appendix A), and per-replicate reports (Appendix A) can be found in the Appendix A. For reference, a comparison between the representative TTMD trajectory for the best (pose1, MD1) and worst (pose4, MD2) scoring pose for complex 4V25, according to the MS coefficient, is highlighted in Video S3.

As depicted in Figure 7, which reports the results for the TTMD post-docking refinement performed on docking poses of ligand SZ6 (PDB ID: 4V25), once again the MS score can identify the native-like docking pose among a set of decoys. Particularly, pose 1 (which is the closest to the reference), pose 2, and pose 5, all share the same correct orientation of the core (spanning from the dihydroxybenzamide moiety to the chloromethylpyrimidine one) within the catalytic site, and present significantly lower MS value compared to poses 3 and 4, where the binding pattern is completely divergent compared to the reference. This can be attributed to the ability of these poses to preserve the pivotal network of direct and water-bridged interactions between the dihydroxybenzamide moiety and residues Asp290, Thr354, and Gly294. As mentioned when discussing some of the previous cases, poses 2 and 5, despite having an incorrect orientation of the difluoroacetylaminomethilbenzyl tail, can easily interconvert throughout the simulation into pose 1 and the crystal reference, because this portion of the molecule is completely solvent-exposed and does not contribute to the interaction with the binding site.

Accordingly, like the case of complex 5MQV, where the core of the molecule was placed correctly in most poses and the difference was mainly related to the tail of the molecule which extended out of the binding site, in this case the MS coefficient was also able to distinguish poses with a high RMSD but native-like interaction pattern from poses with high RMSD but incorrect interaction features.

**Figure 8 ijms-24-03596-f008:**
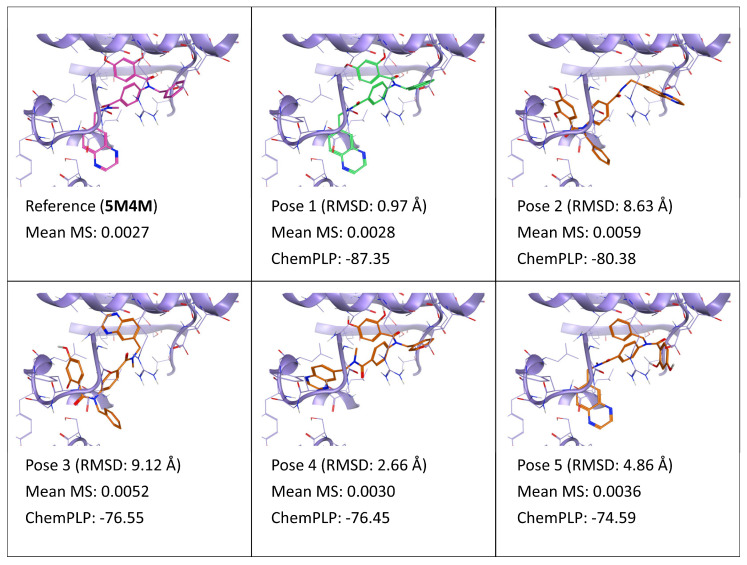
This panel encompasses the results of the TTMD post-docking refinement of poses generated for ligand 7FW within the catalytic site of the pyruvate dehydrogenase kinase 2 (PDB ID: 5M4M). Protein residues within 5 Å of the ligand (orange) position are shown (violet). For each pose, the RMSD in comparison with the reference, the average MS coefficient resulting from TTMD simulations, and the ChemPLP docking score are reported. As a reference, the experimentally determined binding mode (magenta) is also reported, as well as the MS coefficient derived from its TTMD refinement, while the best docking pose (according to the RMSD to the reference) is highlighted in green.

Similar behavior can be observed also for ligand 7FW (PDB ID: 5M4M), as noticeable in Figure 8. In this case, the two native-like poses (poses 1 and 4) outscore the other ones, according to the MS coefficient. In both cases, the ligand’s dihydroxybenzamide moiety is correctly located within a subpocket where it can establish multiple directional interactions with Asp290, Thr54, and Gly294, although in the case of pose 4, the quinoxaline moiety is flipped compared to the reference and pose 1 and interacts with the backbone of Leu330. Interestingly, pose 5 (which occupies the pocket with a similar orientation to poses 1 and 4, but inserts an undecorated phenyl ring within the aforementioned pocket instead of the dihydroxymethoxybenzamide) has a higher but still reasonable MS score, contrary to poses 2 and 3 where the complementarity of shape and electrostatic properties between the ligand and the pocket are inferior, and most of the interaction between the ligand the binding site occurs in the lower portion, the subpocket, which in the reference pose is occupied by the quinoxaline group.

**Figure 9 ijms-24-03596-f009:**
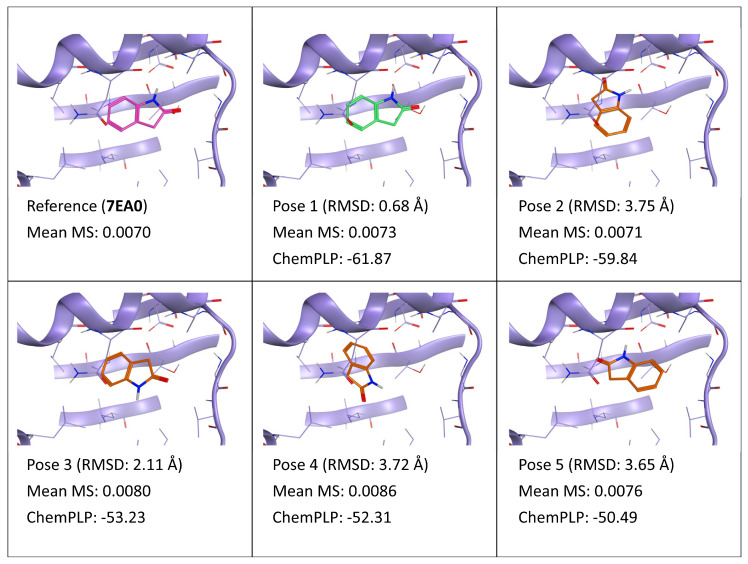
This panel encompasses the results of the TTMD post-docking refinement of poses generated for ligand W6P within the catalytic site of the pyruvate dehydrogenase kinase 2 (PDB ID: 7EA0). Protein residues within 5 Å of the ligand (orange) position are shown (violet). For each pose, the RMSD in comparison with the reference, the average MS coefficient resulting from TTMD simulations, and the ChemPLP docking score are reported. As a reference, the experimentally determined binding mode (magenta) is also reported, as well as the MS coefficient derived from its TTMD refinement, while the best docking pose (according to the RMSD to the reference) is highlighted in green.

Contrary to the first two complexes involving strong binders and coherently with what was observed for complexes 5IH6 and 6HOU, the TTMD refinement was also inconclusive for poses generated for ligand W6P (PDB ID: 7EA0). Once again, the poses were all unstable under the chosen simulation conditions (see also Appendix A, Appendix A, for reference), leading to high MS values and concealment of the subtle differences between different ligand conformation and interaction patterns. The third observance of this phenomenon is a further reaffirmation that the default temperature ramp is too aggressive for elucidating small differences in binding pose stability such as the ones of weak binders.

### 2.4. SARS-CoV-2 Main Protease

The fourth and final test case involved the SARS-CoV-2 main protease (M^pro^), an enzyme playing a pivotal role in this virus replication cycle [46] thanks to its ability to drive the maturation of several nonstructural proteins through the proteolytic cleavage of the pp1a/pp1ab polyproteins [47]. The result of the TTMD refinement carried out on docking poses generated for each M^pro^ protein-ligand complex is summarized in Figure 10, Figure 11 and Figure 12, while detailed analysis of representative replicates (Appendix A), and per-replicate reports (Appendix A) can be found in the Appendix A. For reference, a comparison between the representative TTMD trajectory for the best (pose1, MD5) and worst (pose2, MD4) scoring pose for complex 7M8P, according to the MS coefficient, is highlighted in Video S4.

In the case of ligand YSJ (PDB ID: 7M8P), the MS value determined through the TTMD post-docking refinement indicates pose 1, which is superimposable to the crystal reference, as the most stable pose (see also Video S5, Appendix A). The evident difference between the stability of pose 1 compared to the other ones can be related to the fact that it is the only docking pose that contemporarily satisfies both hydrogen bond interactions with the sidechain of His163 and the backbone of Glu166 (see also Appendix A, Appendix A), which are known to be essential for binding at the catalytic site of the protease [48]. Another possible explanation for the enhanced gap in poses’ stability compared to previously discussed cases is that, in the case of kinases, the binding pocket is usually very druggable, implying that there are several possible ligand conformations with reasonable complementarity of shape and electrostatic properties with the cavity, hence with high stability during MD simulations. On the contrary, M^pro^ catalytic site is very shallow and solvent-exposed [49], implying more constraints on possible ligand orientations that lead to a stable binding. Indeed, short-range nondirectional interactions play a lesser important role in ligand recognition at this site, with more importance being given to a reduced number of high-quality interactions centered upon the S1 subpocket.

**Figure 11 ijms-24-03596-f011:**
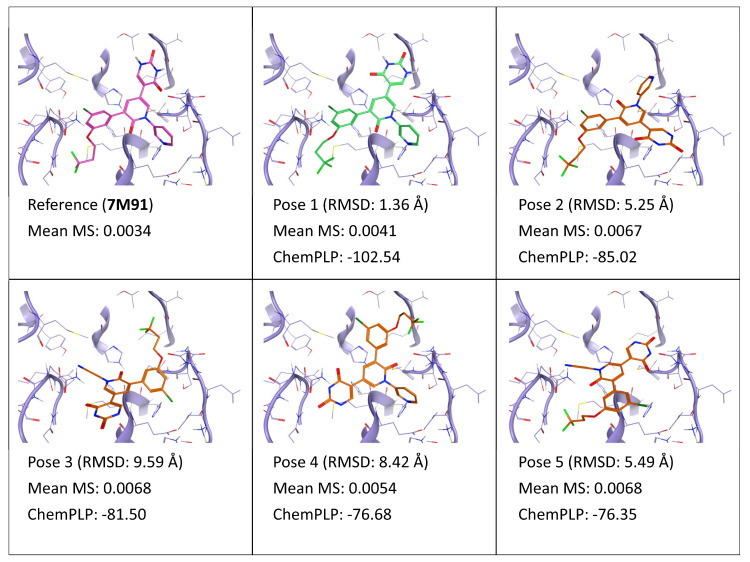
This panel encompasses the results of the TTMD post-docking refinement of poses generated for ligand YU4 within the catalytic site of the SARS-CoV-2 main protease (PDB ID: 7M91). Protein residues within 5 Å of the ligand (orange) position are shown (violet). For each pose, the RMSD in comparison with the reference, the average MS coefficient resulting from TTMD simulations, and the ChemPLP docking score are reported. As a reference, the experimentally determined binding mode (magenta) is also reported, as well as the MS coefficient derived from its TTMD refinement, while the best docking pose (according to the RMSD to the reference) is highlighted in green.

Accordingly, similar results are obtained also for ligand YU4 (PDB ID: 7M91). As depicted in Figure 11, the MS value once again correctly discriminates between the native-like ligand conformation (pose 1) and incorrect binding poses (2–5). Interestingly, pose 4, which presents one of the two conserved hydrogen bonds (the one with His163), is the second most stable pose under simulation conditions, further highlighting the importance of this interaction in anchoring the ligand to the catalytic site. Furthermore, pose 4 is more stable than poses 2, 3, and 5, despite being the only one without a hydrophobic/aromatic ring within the S2 subpocket, another important binding feature of many crystal ligands, including natural peptide substrates [50].

**Figure 12 ijms-24-03596-f012:**
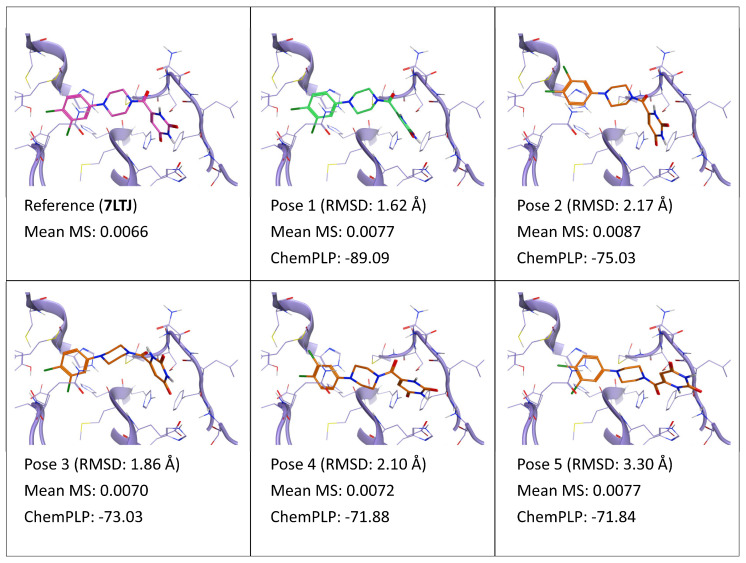
This panel encompasses the results of the TTMD post-docking refinement of poses generated for ligand YD1 within the catalytic site of the SARS-CoV-2 main protease (PDB ID: 7LTJ). Protein residues within 5 Å of the ligand (orange) position are shown (violet). For each pose, the RMSD in comparison with the reference, the average MS coefficient resulting from TTMD simulations, and the ChemPLP docking score are reported. As a reference, the experimentally determined binding mode (magenta) is also reported, as well as the MS coefficient derived from its TTMD refinement, while the best docking pose (according to the RMSD to the reference) is highlighted in green.

Finally, as observed for all previous biological targets, in this case also the TTMD refinement of docking poses generated for the weak binder (ligand YD1, PDB ID: 7LTJ) was inconclusive. The MS values are high and quite similar although expectable due to the similarity among this set of poses and the crystal reference and the low experimentally determined IC_50_ value.

## 3. Discussion

In the present work, we presented the first application of Thermal Titration Molecular Dynamics (TTMD), an alternative enhanced sampling MD-based method for the qualitative estimation of protein-ligand unbinding kinetics, as a post-docking refinement protocol. TTMD, which combines a series of molecular dynamics simulations performed at progressively increasing temperatures with a scoring function based on protein-ligand interaction fingerprints, was used to rank a series of docking poses based on the persistence of the native binding mode. Four different pharmaceutically relevant biological targets, including protein kinase CK1δ, protein kinase CK2, pyruvate dehydrogenase kinase 2, and SARS-CoV-2 main protease were investigated, with three different protein-ligand complexes for each target used in this work. Test cases were chosen among the same protein-ligand complexes used in the original paper [25], as well as the same simulation conditions, i.e., the adopted temperature ramp.

Results reported in this manuscript show how TTMD can be readily used as a post-docking refinement tool for mature, drug-like ligands. The results of the TTMD refinement performed on docking poses generated for this class of ligands showed how the MS score was consistently able to identify the native-like binding pose among the other ones. Furthermore, the MS score demonstrated a good sensitivity in discriminating between poses with partially overlapping interaction features, which could be tricky to evaluate based only on docking calculations.

Most MD-based post-docking refinement protocols involve performing short simulations (tens of nanoseconds) at a fixed temperature to distinguish between stable and unstable poses. The discrimination is usually based on simple geometric descriptors such as the ligand RMSD or the center-of-mass distance between the ligand and the protein. Although this strategy seems legitimate and could work in some cases, based on our experience and as can be observed by the detailed trajectory analyses provided in the Appendix A, these types of simulations are rarely successful for this purpose. First, short windows of classic MD simulations are not sufficient to appreciate significant differences between plausible, near-native poses and the native binding mode, since they probably satisfy a good number of interactions with the target allowing them to occupy the binding site for a sufficient time to be considered stable, even if incorrect. Second, there is no clear cutoff to use for a geometric descriptor to define if the starting binding mode is lost or not: depending on the ligand structure and topological properties of the binding site, values would have to be adjusted in a case-by-case scenario. Moreover, geometric descriptors do not discriminate well if the deviation from the starting point of the simulation regards the whole ligand or only a portion of it: in some cases, as shown in the Results section, some ligand moieties do not interact with the protein active site and are therefore relatively unconstrained. These ligand chunks contribute to increasing the ligand RMSD without causing a loss of interaction with the target, falsely leading to a perception of pose instability.

On the contrary, thanks to its simple but effective enhanced sampling strategy based on increasing the temperature in between each simulation window, TTMD can emphasize stability differences between relatively similar or partially overlapping docking poses in simulation times comparable to those of classical post-docking refinements (see Appendix A for statistics about calculation speed on each investigated system). Moreover, the use of the fingerprint-based scoring function (IFP_CS_) instead of the ligand RMSD allows for determining a clear cutoff value (zero) that can univocally define whether the loss of interactions found in the starting protein-ligand complex happened. This threshold value is used as an early termination criterion, avoiding spending time simulating the ligand floating around the simulation box thus increasing the feasibility of using such an approach in a high-throughput fashion. Indeed, as can be deduced by Appendix A (Appendix A), several simulations end with the total detachment of the ligand from the binding site (considerably high RMSD values). This fact has two main consequences: first, two poses that detach at different times/temperatures would be considered equally unstable if the final/average RMSD would be used as a metric. Second, one would have to simulate for a long time even in those cases where it is not necessary.

Although the TTMD protocol can be useful as is, two questions remain unaddressed still. The first issue with the current protocol is related to the applicability to weak binders, i.e., compounds with affinity values that are above the micromolar threshold, including fragments, which are the starting point of many drug discovery campaigns these days. As reported in the Results section of the manuscript, the current temperature ramp is too aggressive to be applied to weakly binding ligands. Particularly, the aggressiveness of the ramp causes both native-like and wrong poses to rapidly lose their binding features and eventually detach from the binding site. This situation could be addressed by using an alternative ramp, focused on lower temperature ranges. Furthermore, as highlighted by Appendix A (Appendix A) in the case of complex 5IH6, but also in Video S5 (Appendix A) which regards complex 7M8P, it is possible that poses interconvert between each other during the TTMD simulation, especially for low-affinity compounds such as fragments. To better keep track of this behavior, an alternative analysis protocol could be to analyze the frequency by which a certain pose is sampled across the whole set of replicates for a given ligand, regardless of the starting ligand conformation. For both these latest aspects, work in this direction is currently being carried out in our laboratory and will be the scope of a future paper. The second problem is to make the protocol suitable also for investigating membrane proteins. The question is not trivial, since these types of targets are difficult to simulate accurately in standard conditions and therefore deserve special care when dealing with unusual simulation conditions such as the high temperatures explored with the current protocol. A possible solution, which is currently being evaluated in our laboratory, is to simulate these proteins as soluble, without introducing variables related to the membrane behavior.

## 4. Materials and Methods

### 4.1. Hardware Overview

Most of the molecular modeling tasks, like the preparation of protein and ligand structures, docking calculations, the setup for MD simulations, and MD trajectory analyses, were carried out on a Linux workstation, equipped with a 20 cores Intel Core i9-9820X 3.3 GHz processor, running Ubuntu 20.04 as its operating system. An in-house GPU cluster, composed of 20 NVIDIA drivers ranging from GTX980 to RTX2080Ti, has been exploited for carrying out each MD simulation.

### 4.2. Structure Preparation

The three-dimensional coordinates of each protein-ligand complex used in this study were retrieved from the Protein Data Bank (PDB) [51] and processed through various tools embedded within the Molecular Operating Environment (MOE) 2022.02 suite [52]. In the present work, four different biological targets were considered: casein kinase 1 isoform δ (CK1δ), casein kinase 2 (CK2), pyruvate dehydrogenase kinase 2 (PDK2), and SARS-CoV-2 main protease (M^pro^). For each macromolecule, three different protein-ligand complex structures were considered: the accession codes for each structure are summarized in Table 1.

Protein-ligand systems were simulated as monomers, aside from SARS-CoV-2 M^pro^, for which the catalytically competent homodimer form was reconstituted through symmetric crystallographic transformations. Initially, each structure was pre-processed through the “structure preparation” tool, to assign residues with alternative conformations to the highest occupancy one, to rebuild broken loops through homology modeling, and to fix inconsistencies between the aminoacidic sequence and the experimentally solved structure. Afterward, missing hydrogen atoms were added to the system according to the most probable protonation and tautomeric state at pH = 7.4 through the “Protonate3D” tool. Lastly, each solvent, co-solvent, and ion atom was removed before storing the prepared complex for subsequent calculations.

### 4.3. Docking Calculations

For each protein-ligand complex, a self-docking experiment was conducted. Each ligand was prepared using the “Wash” tool of the MOE 2022.02 suite, assigning the most probable protomeric/tautomeric state at pH = 7.4, rebuilding its three-dimensional coordinates, and attributing the correct partial charges, according to the MMFF94 force field [53]. The binding site was defined using a sphere of radius 10.5 Å centered upon the center of mass of the co-crystallized ligand in the experimentally solved structure. Docking calculations were carried out using the Protein-Ligand ANT System (PLANTS) [54,55] program, which uses an Ant Colony Optimization (ACO) search algorithm and is free to use for academics. The first five poses according to the ChemPLP [56] scoring function were stored for further calculations.

### 4.4. System Setup for MD Simulations and Equilibration Protocol

Each protein-ligand complex obtained through docking calculations was prepared for MD simulations through the combination of different packages included in both Visual Molecular Dynamics (VMD [57]) 1.9.2 and the AmberTools22 [58,59] suite. Parameters for protein, water molecules, and ions were attributed according to the ff14SB [60] force field, while ligands were parametrized utilizing the general Amber force field (GAFF [61]). Ligand partial charges were calculated through the AM1-BCC [62] method. Each protein-ligand system was solvated within a rectangular base prism, ensuring a 15 Å padding value between the box border and the nearest protein atom and using the TIP3P [63] model for water molecules. The proper number of monovalent ions, i.e., sodium and chlorine, were then added to neutralize the net charge of the box and to reach a salt concentration of 0.154 M. The prepared system was then subjected to 500 steps of energy minimization with the conjugate-gradient algorithm before undergoing MD simulations, to remove clashes and bad contacts.

Each MD simulation presented in this work was conducted using the ACEMD [64] 3.5 engine, which is based upon OpenMM [65], an open-source Python library for molecular simulations. Both in the equilibration and the production stage, an integration timestep of 2 fs was used, the length of bonds involving hydrogen atoms was constrained through the M-SHAKE algorithm, electrostatic interactions were computed through the particle-mesh Ewald [66] method, using cubic spline interpolation and a 1 Å grid spacing, while a 9.0 Å cutoff was used for calculating Lennard–Jones interactions.

Before productive MD simulations, each system was equilibrated in a two-step process. In the first equilibration run, a 0.1 ns simulation in the canonical ensemble was carried out, imposing harmonic positional restraints on both the protein and ligand atoms, leaving the water molecules and the ions unconstrained. In the second equilibration run, a 0.5 ns simulation in the isothermal-isobaric ensemble was performed, with harmonic positional restraints applied only on the protein backbone and on the ligand atoms, leaving the protein sidechain unconstrained. In both cases, a force constant of 5 kcal mol^−1^ Å^−2^ was applied for the whole duration of each equilibration run on each restrained atom. For both equilibration stages, the temperature was kept at a constant 300 K value through a Langevin thermostat [67], while in the second stage, the pressure was maintained at the constant value of 1 atm through a Monte Carlo barostat [68].

### 4.5. Thermal Titration Molecular Dynamics (TTMD) Simulations

As thoroughly described in the work of Pavan et al. [25], Thermal Titration Molecular Dynamics (TTMD) is an alternative enhanced sampling MD approach originally developed for the qualitative estimation of protein-ligand unbinding kinetics. The sampling method consists of a series of short classic MD simulations (defined as TTMD-steps), performed at progressively increasing temperature values. The progress of the simulation is monitored by evaluating the conservation of the starting binding mode for a certain protein-ligand complex (a docking pose, in this case) through an interaction fingerprint-based scoring function [26], originally implemented in open-source Autogrow4 [69] program, a semi-automatized workflow for de novo drug design. The protocol described afterward is implemented as a Python 3.10 code and relies on the NumPy, MDAnalysis [70,71], Open Drug Discovery Toolkit [72], and Scikit-learn external libraries. The code to run TTMD simulations is available free of charge, under a permissive MIT license, at github.com/molecularmodelingsection/TTMD (accessed on 25 January 2023).

The user must define the “temperature ramp”, i.e., the temperature values that are sampled throughout the simulation and the time spent simulating at each temperature. Different from the original version of the code, in the current one, the length of each simulation window is not fixed, so different temperatures can be sampled for a different amount of time. In the present work, for consistency with the previous paper, the same temperature ramp was used: the starting temperature was set at 300 K, the end temperature was set at 450 K, the temperature increment between each TTMD-step was set at 10 K, while the length of each TTMD-step was 10 ns. The extension of the temperature ramp must be carefully chosen based on prior knowledge of the target, especially concerning the conservation of the native protein fold throughout the simulation (which, in this case, was carried out by monitoring the backbone RMSD).

As anticipated, the progress of the simulation is evaluated by a scoring function based on protein-ligand interaction fingerprints. This scoring protocol, named IFP_CS_, exploits the Scikit-learn Python module to calculate the cosine similarity between two vectors, i.e., the protein-ligand interaction fingerprints of a reference and a query, where the reference is the protein-ligand complex extracted from the last frame of the second equilibration stage, and the query is each protein-ligand complex obtained from each frame of a certain TTMD-step. Specifically, each interaction fingerprint is an integer vector of r × 8 length, where r represents the number of protein residues and 8 represents the possible type of protein-ligand interaction that can be encoded in the vector (hydrophobic contacts, aromatic face-to-face, aromatic edge-to-face, hydrogen bonds with the protein acting as a donor, hydrogen bonds with the protein acting as an acceptor, salt bridge with the protein acting as the positively charged member, salt bridge with the protein acting as the positively negative member, and an ionic bond with a metal ion, respectively). The cosine similarity value is then multiplied by −1 to comply with most scoring functions, where negative values indicate high affinity towards the target and low, near-zero, values indicate lower affinity.

The values of IFP_CS_ can range from −1, highlighting a total convergence between the reference (the native binding mode) and the query (the binding pose sampled at a certain time during the simulation), to zero, indicating that each interaction feature of the reference has been lost during the simulation.

After the conclusion of each TTMD-step, the average IFP_CS_ calculated across each frame of the step is calculated: if this value is null, implying that the native binding mode has not been sampled for the whole duration of the step, the TTMD simulation is terminated; otherwise, the simulation proceeds with the next TTMD-step.

### 4.6. Trajectory Analyses and MS Coefficient Determination

TTMD trajectories were analyzed through an in-house Python script. Specifically, the RMSD of the protein backbone and the ligand are calculated at each frame by exploiting the appropriate function provided by the MDAnalysis package. A time-dependent per-residue decomposition of the protein-ligand interaction energy is calculated by exploiting the NAMD Energy plugin 1.4 [73] for VMD. Afterward, three analysis plots are generated by exploiting the Matplotlib and Seaborn Python packages.

The first plot, defined as the “titration profile”, correlates the average IFP_CS_ value with the temperature of each step: the slope of the straight line linking the initial and the final state of the titration is defined as the MS coefficient and is used as a proxy measure for estimating the “residence time” of the investigated pose.
(1)$$\mathrm{MS}=\frac{meanIFP_{CS}^{Tend}-\left(-1\right)}{T^{end}-T^{start}}$$

The MS coefficient ranges from zero (indicative of a tight binding/high persistence of the native binding mode) to 1 (which, on the contrary, indicates low persistence of the starting binding pose). For each investigated protein-ligand complex, five independent TTMD simulations are carried out, with the average MS coefficient being calculated across three different replicates, after discarding the highest and the lowest value.

The second plot reports the time-dependent per-residue decomposition of the protein-ligand interaction energy for the 25 protein residues that are most frequently contacted by the ligand throughout the trajectory (using a distance cutoff of 4.5 Å). The third and final plot reports the time-dependent evolution of the ligand and protein backbone RMSD and the IFP_CS_ score.

## Figures and Tables

**Figure 1 ijms-24-03596-f001:**
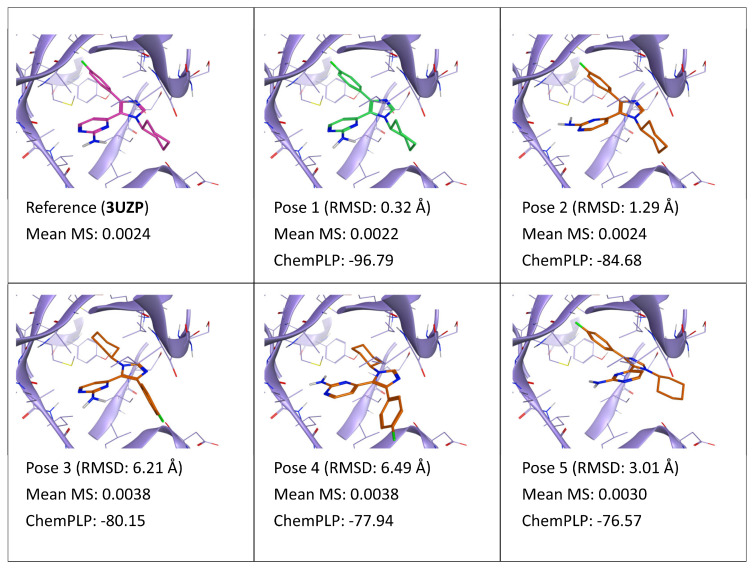
This panel encompasses the results of the TTMD post-docking refinement of poses generated for ligand 0CK within the catalytic site of the protein kinase CK1δ (PDB ID: 3UZP). Protein residues within 5 Å of the ligand (orange) position are shown (violet). For each pose, the RMSD in comparison with the reference, the average MS coefficient resulting from TTMD simulations, and the ChemPLP docking score are reported. As a reference, the experimentally determined binding mode (magenta) is also reported, as well as the MS coefficient derived from its TTMD refinement, while the best docking pose (according to the RMSD to the reference) is highlighted in green.

**Figure 4 ijms-24-03596-f004:**
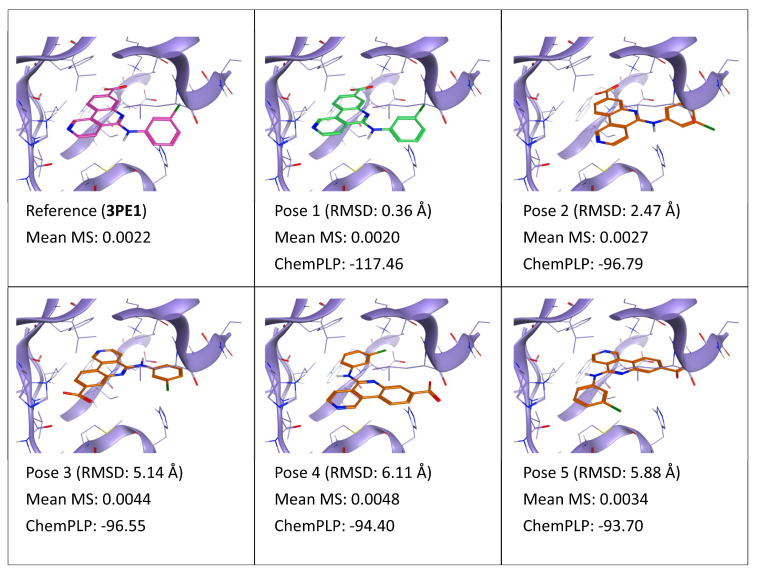
This panel encompasses the results of the TTMD post-docking refinement of poses generated for ligand 3NG within the catalytic site of the protein kinase CK2 (PDB ID: 3PE1). Protein residues within 5 Å of the ligand (orange) position are shown (violet). For each pose, the RMSD in comparison with the reference, the average MS coefficient resulting from TTMD simulations, and the ChemPLP docking score are reported. As a reference, the experimentally determined binding mode (magenta) is also reported, as well as the MS coefficient derived from its TTMD refinement, while the best docking pose (according to the RMSD to the reference) is highlighted in green.

**Figure 7 ijms-24-03596-f007:**
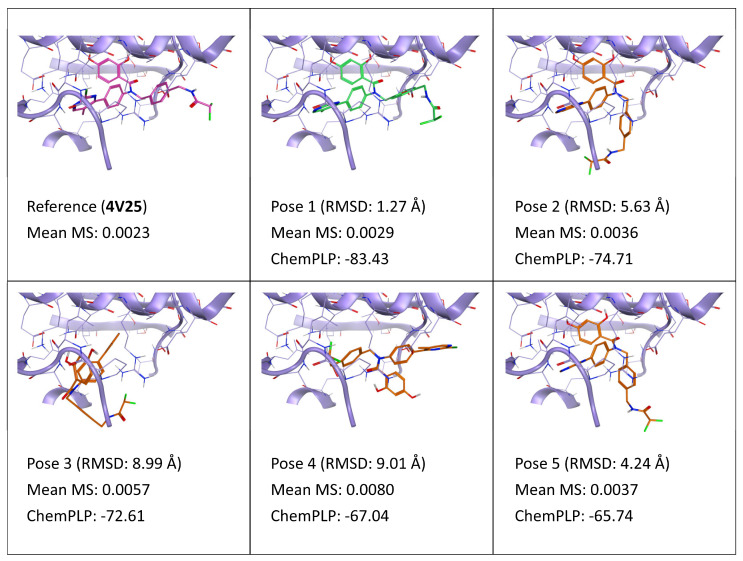
This panel encompasses the results of the TTMD post-docking refinement of poses generated for ligand SZ6 within the catalytic site of the pyruvate dehydrogenase kinase 2 (PDB ID: 4V25). Protein residues within 5 Å of the ligand (orange) position are shown (violet). For each pose, the RMSD in comparison with the reference, the average MS coefficient resulting from TTMD simulations, and the ChemPLP docking score are reported. As a reference, the experimentally determined binding mode (magenta) is also reported, as well as the MS coefficient derived from its TTMD refinement, while the best docking pose (according to the RMSD to the reference) is highlighted in green.

**Figure 10 ijms-24-03596-f010:**
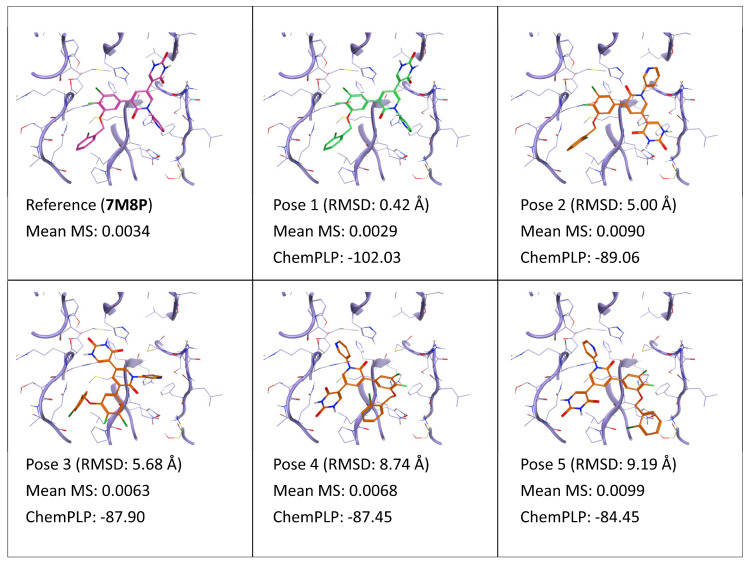
This panel encompasses the results of the TTMD post-docking refinement of poses generated for ligand YSJ within the catalytic site of the SARS-CoV-2 main protease (PDB ID: 7M8P). Protein residues within 5 Å of the ligand (orange) position are shown (violet). For each pose, the RMSD in comparison with the reference, the average MS coefficient resulting from TTMD simulations, and the ChemPLP docking score are reported. As a reference, the experimentally determined binding mode (magenta) is also reported, as well as the MS coefficient derived from its TTMD refinement, while the best docking pose (according to the RMSD to the reference) is highlighted in green.

**Table 1 ijms-24-03596-t001:** PDB accession codes for protein-ligand complex structures employed in this work ^a^.

CK1δ	3UZP [27]	5IH6 [28]	5MQV [29]
CK2	3PE2 [30]	6HOU [31]	3PE1 [30]
PDK2	4V25 [32]	7EA0 [33]	5M4M [34]
M^pro^	7LTJ [35]	7M91 [36]	7M8P [36]

^a^ Structures are grouped by the target.

## Data Availability

All molecular structures utilized in this work have been retrieved from the publicly available PDB (www.rcsb.org (accessed on 25 January 2023)). The TTMD.py Python code to reproduce the simulations performed in this work, as well as a YAML file to reconstitute the appropriate virtual environment to run them, is available at github.com/molecularmodelingsection/TTMD (accessed on 25 January 2023) and released under a permissive MIT license. The TTMD.py script carries on each step of a TTMD simulation, including the system setup and parametrization, the equilibration protocol, the production runs, and associated analyses.

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
