# Peer review of "Thermal Titration Molecular Dynamics (TTMD): Not Your Usual Post-Docking Refinement"

_ijms, 2023, doi:10.3390/ijms24043596_

Round 1
Reviewer 1 Report
This work is a nice research which definitely has its merits in the field of computational drug design. There are however some remarks of items that must be improved:
1) There should be a space between text and reference links (e.g. [1])
2) Fractional numbers should be written with a point (.) as fractional indicator (instead of comma)
3) The references should be carefully inspected for missing spaces
4) Please add a table that summarises the compounds in terms of their biological affinities related to the corresponding target (i.e. add a table with pIC50/Kd values), since now the reader cannot distinguish between strong and weak binders in the conclusions.
5) Figures 1 to 12 are hard to read and not meaningful for the discussion. It would make it much more readable if a plot would be included that correlates the pIC50/Kd values with the obtained "mean MS" values. Figure 1-12 can be moved to the SI.
6) For each target, three compounds were evaluated (1 weak binder and 2 strong binders). It would make the research more convincing if cross-evaluations were added: for each protein target, also repeat the docking/TTMD simulations with the 9 ligands from the other 3 targets. This would increase the total number of ligands per target from 3 (current work) to 12 ligands per target, hence showing that the methodology can be applied to ligands that are false positives (according the docking). It would also indicate that the technology can be "automated" (which is often essential in nowadays computational drug design).
Author Response
This work is a nice research which definitely has its merits in the field of computational drug design. There are however some remarks of items that must be improved:
1) There should be a space between text and reference links (e.g. [1])
2) Fractional numbers should be written with a point (.) as fractional indicator (instead of comma)
3) The references should be carefully inspected for missing spaces
4) Please add a table that summarises the compounds in terms of their biological affinities related to the corresponding target (i.e. add a table with pIC50/Kd values), since now the reader cannot distinguish between strong and weak binders in the conclusions.
5) Figures 1 to 12 are hard to read and not meaningful for the discussion. It would make it much more readable if a plot would be included that correlates the pIC50/Kd values with the obtained "mean MS" values. Figure 1-12 can be moved to the SI.
6) For each target, three compounds were evaluated (1 weak binder and 2 strong binders). It would make the research more convincing if cross-evaluations were added: for each protein target, also repeat the docking/TTMD simulations with the 9 ligands from the other 3 targets. This would increase the total number of ligands per target from 3 (current work) to 12 ligands per target, hence showing that the methodology can be applied to ligands that are false positives (according the docking). It would also indicate that the technology can be "automated" (which is often essential in nowadays computational drug design).
We thank the reviewer for the positive feedback on our work. Here is our point-by-point response to the issues raised during the reviewing phase.
- The issue was addressed in the manuscript.
- The images and tables reporting the wrong spacer characters were corrected.
- The references were checked and should now be correct.
- We thank the reviewer for the suggestion, which will make the manuscript more comprehensive. We initially omitted the requested data since they were already reported in the original publication reporting TTMD as a new methodology, which was referenced throughout the manuscript, but we agree that the reader would benefit from having this information included in the paper. We added the requested data in Tables S1, S4, S7, and S10.
- The reviewer is raising an interesting point, which is however beyond the scope of this manuscript. The usage of TTMD as a tool to classify ligands based on their target affinity has already been discussed in the original paper concerning the TTMD methodology by Pavan et al. (https://doi.org/10.1021/acs.jcim.2c00995), where plotsofKd/IC50versus MSare reported (Figures 2,3,4,5).In the present article, instead, we used TTMD as a docking pose selector, assessing if the MS score could be helpful to discriminate between different poses of the same ligand and to identify the native-like ones. Figures 1-12 report the main results of the article and illustrate the data as the basis of the discussion: for this reason, we think that these figures belong to the main text and not to Supplementary Information. Accordingly, since we are comparing different docking poses of the same ligand, emphasizing the correlation between the MS score and the activity/affinity value of different ligands would deviate from the goal of the manuscript. We mentioned the difference between strong and weak binders just to define the applicability range of the technique, which works in the case of strong binders while requiring the development of a milder ramp for weak binders.
- Likewise to what was reported for point 5, comparing poses of different ligands and discriminating between them is beyond the scope of the manuscript, which is uniquely focused on using TTMD as an alternative scoring protocol to rank docking poses of the same ligand. The capability of TTMD to distinguish high-affinity from low-affinity ligands was accessed in the aforementioned publication (https://doi.org/10.1021/acs.jcim.2c00995) and will be further investigated in the future. We do not think that adding the suggested analysis here would be beneficial, given the purpose of the manuscript, but we are open to suggestions on how to improve the paper’s readability to avoid any misunderstanding.
Reviewer 2 Report
In the article entitled “Thermal Titration Molecular Dynamics (TTMD): not your usual 2 post-docking refinement” the authors present a method for qualitative scoring of docking poses by using molecular dynamics. The article is very interesting and well written and only needs a few minor corrections before publication. My comments are as follows:
1. What is the advantage of the TTMD approach in comparison with standard MD simulations. As can be seen from the manuscript, verification of a single docking pose required 5 MD simulations of 160 ns (which is at least 5 days of GPU time). This is in contradiction with the “high throughput” purpose of docking. The idea itself is very interesting and valid, but can the authors describe a practical benefit of using their method compared to using standard MD simulations with e.g. MM/GBSA or MM/PBSA method?
2. Additionally, have the authors considered how would they approach validation of docking poses for thermally less stable proteins since in their method they increase the temperature up to 450 K (177 °C), which is even well above the boiling temperature of water. Did this cause any problems in the simulations?
3. The MS value needs to be defined in the main text
4. In line 178 page 5, the authors mention “a triple hydrogen bond”. Do they mean three hydrogen bonds?
5. In line 225 page 7, words “whole” and “naphthyridine” need to be separated.
6. In SI figures (e.g., Fig S1), symbols “A”, “B”, … are missing from the figure.
7. Additionally, in Figure description (e.g., Fig S1), word “line” is missing in the sentence “with the slope of the straight being reported in the legend”
Author Response
In the article entitled “Thermal Titration Molecular Dynamics (TTMD): not your usual 2 post-docking refinement” the authors present a method for qualitative scoring of docking poses by using molecular dynamics. The article is very interesting and well written and only needs a few minor corrections before publication. My comments are as follows:
- What is the advantage of the TTMD approach in comparison with standard MD simulations. As can be seen from the manuscript, verification of a single docking pose required 5 MD simulations of 160 ns (which is at least 5 days of GPU time). This is in contradiction with the “high throughput” purpose of docking. The idea itself is very interesting and valid, but can the authors describe a practical benefit of using their method compared to using standard MD simulations with e.g. MM/GBSA or MM/PBSA method?
- Additionally, have the authors considered how would they approach validation of docking poses for thermally less stable proteins since in their method they increase the temperature up to 450 K (177 °C), which is even well above the boiling temperature of water. Did this cause any problems in the simulations?
- The MS value needs to be defined in the main text
- In line 178 page 5, the authors mention “a triple hydrogen bond”. Do they mean three hydrogen bonds?
- In line 225 page 7, words “whole” and “naphthyridine” need to be separated.
- In SI figures (e.g., Fig S1), symbols “A”, “B”, … are missing from the figure.
- Additionally, in Figure description (e.g., Fig S1), word “line” is missing in the sentence “with the slope of the straight being reported in the legend”
We thank the reviewer for the positive feedback on our work. Here are responses to the issues raised during the reviewing phase.
- Based on our experience, a normal post-docking MD simulation is less sensitive than TTMD in discriminating between the top-ranked docking solutions, since they usually all provide a good set of interactive features with the binding pocket. As reported in the analysis panels in Supplementary Materials, short simulation times and standard simulation temperatures produce little to no difference in stability between different poses of the same ligand. It is only by reaching higher temperatures that those differences in stability can be appreciated. Regarding the simulation time, one could develop custom temperature ramps that are shorter and centered upon a certain temperature interval that is good for the target of interest. In this case, we were interested in applying the method as is, since this facilitates the work of non-experts and increases the possibility of adopting the method for a wider variety of targets, but it is certainly optimizable according to different necessities by each laboratory. Regarding the scalability of this approach, MD simulations in general are time-consuming due to the limitations of today's hardware, but the rapid increase in simulation speed stemming from the development of novel, more powerful, GPUs, points at a near future where this type of calculations will be much more affordable. Nevertheless, at the present date, on a modest GPU cluster like the one that we have at our disposal, the complete set of simulations that is required to restore docking poses for a certain compound on a given target, requires a couple of days (at maximum). It does not apply to a high number of ligands, so it must be used in those situations where the result’s accuracy is more important than the speed of execution. Concerning the comparison with MMGBSA/MMPBSA, although it is a good tool for most use cases (as stated in the introduction), it has some limitations that make the result not always reliable, depending on the electrostatic features of the ligand and the binding site and depending on the importance of water molecules in bridging interactions between the ligand and the pocket. In this sense, having an alternative, complementary method such as TTMD would help in those cases where MMGBSA fails.
- Although the temperature used for TTMD simulations indeed exceeds the boiling point of water, the meaning of temperature is not directly translatable to what we can observe in real life. The temperature change only reflects a change in the kinetic energy of particles and not any change of state. This reflects in the good stability of proteins even at high simulation temperatures, as can be observed in the backbone RMSD plots reported in the Supplementary Materials. For the moment, we avoided applying the method to more thermosensitive systems, but we are working on an adjusted method that can be generalized to every target. This is not unheard of, since other similar methods such as scaled MD faced the same issue during their development and came up with a solution to the problem at a later stage.
- The MS coefficient definition has been added to the text (equation 1, line 611, page 18)
- The issue has been addressed in the manuscript.
- The issue has been addressed in the manuscript.
- Panels letters have been added.
- The description has been corrected.
Round 2
Reviewer 1 Report
-